# Multistability in Macrophage Activation Pathways and Metabolic Implications

**DOI:** 10.3390/cells11030404

**Published:** 2022-01-25

**Authors:** Carsten Geiß, Elvira Salas, Jose Guevara-Coto, Anne Régnier-Vigouroux, Rodrigo A. Mora-Rodríguez

**Affiliations:** 1Institute for Developmental Biology and Neurobiology (IDN), Johannes Gutenberg University, 55128 Mainz, Germany; vigouroux@uni-mainz.de; 2Department of Biochemistry, Faculty of Medicine, Campus Rodrigo Facio, University of Costa Rica, San José 11501-2060, Costa Rica; elvira.salas@ucr.ac.cr; 3Department of Computer Sciences and Informatics (ECCI), Faculty of Engineering, Campus Rodrigo Facio, University of Costa Rica, San José 11501-2060, Costa Rica; jose.guevaracoto@ucr.ac.cr; 4Research Center for Information and Communication Technologies (CITIC), Campus Rodrigo Facio, University of Costa Rica, San José 11501-2060, Costa Rica; 5Research Center on Surgery and Cancer (CICICA), Campus Rodrigo Facio, University of Costa Rica, San José 11501-2060, Costa Rica; 6Research Center for Tropical Diseases (CIET), Lab of Tumor Chemosensitivity (LQT), Faculty of Microbiology, Campus Rodrigo Facio, University of Costa Rica, San José 11501-2060, Costa Rica

**Keywords:** macrophage, bistability, multistability, metabolism, systems biology, miRNA

## Abstract

Macrophages are innate immune cells with a dynamic range of reversible activation states including the classical pro-inflammatory (M1) and alternative anti-inflammatory (M2) states. Deciphering how macrophages regulate their transition from one state to the other is key for a deeper understanding of inflammatory diseases and relevant therapies. Common regulatory motifs reported for macrophage transitions, such as positive or double-negative feedback loops, exhibit a switchlike behavior, suggesting the bistability of the system. In this review, we explore the evidence for multistability (including bistability) in macrophage activation pathways at four molecular levels. First, a decision-making module in signal transduction includes mutual inhibitory interactions between M1 (STAT1, NF-KB/p50-p65) and M2 (STAT3, NF-KB/p50-p50) signaling pathways. Second, a switchlike behavior at the gene expression level includes complex network motifs of transcription factors and miRNAs. Third, these changes impact metabolic gene expression, leading to switches in energy production, NADPH and ROS production, TCA cycle functionality, biosynthesis, and nitrogen metabolism. Fourth, metabolic changes are monitored by metabolic sensors coupled to AMPK and mTOR activity to provide stability by maintaining signals promoting M1 or M2 activation. In conclusion, we identify bistability hubs as promising therapeutic targets for reverting or blocking macrophage transitions through modulation of the metabolic environment.

## 1. Introduction

Macrophages are essential components of the innate immune system with multiple functions in both inhibiting/promoting cell proliferation and tissue repair. Diversity and plasticity are hallmarks of macrophages, whereby the classical pro-inflammatory (hereafter referred to as M1) and alternative anti-inflammatory (referred to as M2) activation profiles are thought of as two extremes of their dynamic changing states. The typical characteristics of M1 macrophages include a high capacity for antigen presentation, the intensified production of interleukin (IL)-12 and IL-23, and the increased production of nitric oxide (NO) and reactive oxygen intermediates. M2-type responses are observed in healing-type circumstances without infections. These responses can be further amplified by IL-4, IL-10, or IL-13. An imbalance in M1/M2 macrophage activation is often associated with various diseases, such as cancer or inflammatory conditions. Therefore, M1/M2 activation is a tightly controlled process entailing a set of signaling pathways and transcriptional and post-transcriptional regulatory networks (for a review see [1]).

There is an ongoing controversy surrounding the definition of the activation programs that macrophages undergo during inflammation, immune response, and trained immunity [1,2,3]. However the emerging consensus says that the activation spectrum is a continuum with many different instances defined by pathways with more than one steady state or attractor [3]. The existence of more than one steady state in a system is called multistability and speaks for switchlike behavior. The latter is indeed observed in several important macrophage activation pathways (see below) and can be modeled using systems biology.

Usually, systems have at least one steady state that acts as an attractor, indicating that a time course of the system will bring the concentrations of the model species to a specific state where all the fluxes (production and degradation) are balanced, establishing a minimum in potential energy. Under these conditions, the rate of change is zero until a perturbation moves the system away from that steady state, increasing the potential energy to move back to that attractor. Therefore, when the perturbation is removed, the system will again be conducted to the same steady state. However, under certain conditions, more than one attractor emerges (local minima in potential energy), establishing a multistable system with several possible steady states [4]. A specific case of multistability is bistability (two attractors), establishing a molecular switch. The state of the system can alternatively move from one attractor to the other by overcoming the potential energy required to get out of the region of attraction of one steady state in order to enter the region of attraction of the other steady state (see Figure 1A). Therefore, despite the existence of several attractors with steady-state properties, the molecular pathways in macrophages treated in vitro can overcome the attraction of a first steady state in the presence of the appropriate signals to continue the activation process over time. 

These molecular switches have been reported to emerge in several important regulatory processes of macrophage activation, dependent on various inflammatory mediators, signaling molecules, and transcription factors (TFs). Frequently, the specialized or polarized T cells (Th1, Th2, and regulatory T cells) play an important role in macrophage activation [5], although there are macrophage activities without T- or B-cell influence [6]. At the cellular level, canonical pathways have been postulated to control macrophage activation, such as the interferon regulatory factor (IRF) or the signal transducer and activator of transcription (STAT) signaling pathway. In response to Th1 cytokines and inflammatory stimuli such as interferons (IFNs) and Toll-like receptor (TLR) signaling, stimulation of STAT-1/STAT-2 and IRF-5 primes activation to M1 cells. In contrast, the IL-4- and IL-13-mediated activation of IRF/STAT (via STAT-6) will skew macrophage functions towards the M2 phenotype, while STAT-3 activation by IL-10 is associated with another anti-inflammatory phenotype [7]. Therefore, the IRF/STAT canonical pathway is a potentially interesting candidate, showing at least three different steady states depending on the initial type of stimulation.

These differences in macrophage functions can also be appreciated at the metabolic level, which indicates that the system undergoes a metabolic switch that has significant implications in macrophage activities and the shaping of the metabolic microenvironment. This interplay between metabolism and immune functions is called immunometabolism [8]. As an example, IL-10 signaling has been shown to play a critical role in controlling inflammatory responses by modulating cellular metabolism in activated macrophages [9]. We hypothesize that these metabolites could be in turn sensed by other cells or macrophages to activate signaling molecules able to maintain the gene expression profiles associated with particular macrophage phenotypes and to shape the immunometabolism in the microenvironment of tumors or chronic inflammatory diseases. 

Macrophages are thus not only essential immune cells but also attractive therapeutic targets. The characterization of molecules associated with the dynamic changes of macrophage activation and a deeper understanding of their interactions are crucial for elucidating the molecular basis of disease progression and for designing novel macrophage-based therapeutic strategies. The identification of robust targets controlling the transitions between given steady states is a big challenge. As mentioned above, the M1/M2 transitions observed in vitro are part of a repertoire (or continuum) of activation states of the macrophage. We know of many important bistable switches across the different levels of the regulation of macrophage activation, establishing thereby an array of switches with many potential configurations. We thus hypothesize that the continuum of macrophage activation states analyzed at the level of the cell population is in fact a heterogeneous mixture of single cells in discrete, well-defined states with different configurations of that array of switches. The identification and characterization of the molecular switches in macrophage activation pathways and their implications for macrophage activation states will help to address the complexity of the system and facilitate the identification of molecules critically involved in macrophage phenotype switches.

We therefore aim to review the current evidence supporting a model of multistability in the molecular pathways related to macrophage transitions and connect this evidence with the differences in the metabolic pathways for different macrophage phenotypes. We finally speculate on how these differences in metabolism contribute to the conformation of attractors that self-perpetuate the stability of macrophage phenotypes and the immune modulation of the microenvironment.

## 2. Multistability, Hysteresis, and Ultrasensitivity

The aim of systems biology is not only to quantify the interactions between molecules, but also to predict and describe (quantitatively and qualitatively) emergent properties on the higher level. Each of the proposed hierarchical levels of a system has certain emergent principles that do not appear in the lower level of the organization, and therefore it is impossible to explain the functioning of a biological system using only a reductionist view of the physicochemical principles of the individual components [10]. Of such emergent properties is multistability (including bistability), which is a crucial feature of dynamical systems and is used in various all-or-none kinds of decision-making processes, leading to more than one self-perpetuating steady state (or attractor) [11]. Under these circumstances, two or more steady states coexist in a given set of experimental conditions. The attractors are represented as valleys in a landscape of potential energy (Figure 1A) [4]. This phenomenon was observed early on in a number of biochemical systems [12,13] and has attracted much attention in recent years. Its importance in biological systems, including cell signaling [14], differentiation [15], and the cell cycle [16], is well-known.

A cell signaling network is assembled from frequently occurring motifs that are building blocks for any network [17]. These network motifs play an important role in the propagation of signals in a network and can influence the sensitivity, robustness, and trade-off of the input–output (I/O) relation in a signaling network. The I/O relation is the information processing required for the detection of the amplitude and duration of the incoming signal to generate an output signal of proper strength and duration for the activation of the effectors that alter subcellular processes. A crucial part of the I/O relation is the existence of bistability, where the output signal can attain any of the two stable-activity states that persist under identical parametric conditions in a switchlike behavior [18]. 

A molecular switch implies the emergent property of bistability in a system. This usually arises in biological systems that contain a positive feedback loop or a mutually inhibitory, double-negative feedback loop. Depending on the parameter values of their interactions, these loops create more than one attractor steady state in the system (Figure 1B). It has been considered that at least one positive feedback loop is a necessary requirement for the existence of multiple steady states [19,20], while other authors state that a negative feedback loop is required for stability and a positive feedback loop is necessary for multistability [21]. Indeed, positive feedback circuits cover a wide class of biologically relevant systems with multiple steady states [22]. Nevertheless, the existence of positive loops is far from being sufficient; a positive feedback loop does not guarantee bistability, and this property has to be quantitatively explored for each particular system in order to confirm the emergence of this property [23]. This switchlike behavior is recognized by bifurcation analysis (or phase plane analysis), namely, equilibrium point analysis, which includes studies related to the changes in the qualitative and quantitative structures of the equilibrium points depending on the changes in the model parameters [24]. At bifurcation points, a system’s behavior may differ qualitatively depending on small changes in the bifurcation parameters (those model parameters that enable the system to switch from one steady state to the other) [25].

Due to this attraction of a steady state, a system with positive feedback loops could produce an actively maintained ‘memory’ of a transient inductive stimulus, known as hysteresis [26]. Hysteresis (Figure 1C) is the phenomenon whereby bistable switching is observed for different stimulus responses and the state of the system depends on its history. The trajectory of the system from the steady state A to the steady state B is different from the trajectory from B to A [27]. This property can enrich the adaptation of organisms extending from bacteria to mammals by storing the cellular memory of past stimuli [28,29]. An example of hysteresis in macrophage activation is observed for the phenomenon of lipopolysaccharide (LPS)-induced tolerance. After LPS stimulation, macrophages exhibit memory-like features at the molecular level related to gene-specific chromatin modifications, the silencing of inflammatory molecules, and priming for other genes, modulating the responsiveness of macrophages to subsequent activation [30]. Another example of hysteresis is trained immunity, as it represents a modified steady state of innate immunity after infection [2].

The significance of bistability for the identification of robust therapeutic targets has already been explored. For instance, bistability could lead to another emerging property called ultrasensitivity, where a short perturbation (pulse input) of a signaling cascade leads to a change in a steady state that is self-perpetuated by the system (Figure 1D). This means that the perturbation is strong enough to move the system from the attraction zone of one steady state to the attraction zone of another steady state. This has been described for the mitogen-activated protein kinase (MAPK) cascade that is positively regulated by the activated MAPK [31] and the self-perpetuated activation mechanisms for extracellular signal-regulated kinase (ERK) 1/2, related to bistability [32]. Halder et al. used global sensitivity analysis to identify sensitive parameters and their role in maintaining bistability; they also used bistable switching to explore the underlying principles of the motifs exhibiting bistability [18]. They applied their results for the motifs of protein–protein interactions to identify potential drug targets in cancer networks with potential ultrasensitivity. 

The hysteresis of a bistable switch can be reversible or irreversible depending on the strength of the feedback parameter [15]. A bistable system can reverse back to its previous state if the system shows reversible hysteresis, which is not possible for irreversible hysteresis. In addition, an irreversible hysteresis can be changed to a reversible hysteresis by decreasing the strength of the feedback parameter, suggesting that the manipulation of bistable switches is possible. Halder et al. hypothesize that the circuits with a larger reversible hysteretic range could be better drug targets to make the system switch to a disease-free (OFF) state from a diseased (ON) state by reversing the input parameter, which is more difficult for irreversible hysteresis. They observed that proteins present in the motifs with a higher reversible feedback range tend to be associated with higher numbers of drugs, leading to the identification of ultrasensitive targets. One of the two targets of the drug lapatinib is the epidermal growth factor receptor (EGFR), which belongs to the bistable motif with a higher reversible range [18]. This suggests that the identification of bistable motifs with a high range of reversible hysteresis holds promise for the identification of ultrasensitive targets. These targets would be able to switch the system from a pathological steady state to a resolving steady state.

Multistability in biological systems emerges from the nonlinear behavior of quantitative interactions between the individual components of key molecular pathways (those related to macrophage activation). These pathways have several alternative steady states that work as attractors, robust to the effects of small perturbations. However, when a perturbation is strong enough, it will make the system swap its configuration towards another attractor or steady state. In this sense, a perturbation is self-perpetuated, leading to a switch in the activation of a signaling circuit. Therefore, the identification of ultrasensitive targets to control bistable switches represents the most robust type of therapeutic intervention possible for a system, since the correct use of a pulse signal can switch a system to a different self-perpetuating steady state. Figure 1 illustrates the concepts of multistability, hysteresis, and ultrasensitivity applied to macrophage activation. 

## 3. Current Evidence Pointing to Multistability in Macrophage Programs

IRF/STAT signaling is a central pathway in controlling macrophage M1/M2 activation with the potential for the emergence of multistability. The M1 regulators STAT1, STAT5, IRF5, SOCS3, NFKB-p50-p65, and HIF- 1α have antagonistic and counteracting interactions with the M2 regulators IRF4, SOCS1, STAT3, STAT6, NFKB-p50-p50, and HIF-2α. Several of these transcription factors, including NF-κB, AP-1, and STAT family members, also potentially participate in macrophage trained immunity, resulting in hundreds of times higher gene expression in a short window of time [1,2,7,33].

All these counteracting activities or inhibitory cross-talks between the M1/M2 pathways indicate that the M1/M2 transitions may be considered bistable, as multistable systems consisting of molecular switches of mutually exclusive regulations. To quantitatively decode the underlying principles governing macrophage phenotypic activation and to harness its therapeutic potential in human diseases, a systems-level approach is needed. This is due to the significant number of signaling pathways and intracellular regulatory networks involved. 

The M1/M2 transitions have already been studied using mathematical modeling. Zhao et al. published a mechanistic integrative computational model based on the literature-data-driven description of macrophage activation. They calibrated their ‘virtual macrophage’ model against experimental data, and mechanistically elucidated several signature feedbacks behind the M1/M2 antagonism and investigated the dynamical shaping of macrophage phenotypes within the M1/M2 spectrum. Model sensitivity analysis also revealed key molecular nodes and interactions as targets with potential therapeutic value for the pathophysiology of peripheral arterial disease and cancer [33]. However, they did not perform stability analysis or look for multistability in their model.

The emergent property of bistability has already been reported for macrophage-related pathways and has been studied using mathematical models, such as that of the tyrosine protein kinase (JAK)/STAT signaling pathway. Berez et al. examined how the interaction of STAT, APT (apontic), and SLBO (slow border cells) creates bistability in the JAK/STAT signaling pathway using parameter bifurcation and phase portrait analyses, leading to a model reduction to identify a minimal three-variable quantitative model [34]. They started with a 15-variable model but performed an elegant model simplification to obtain a minimal motif of 3 variables able to present bistability in *Drosophila melanogaster*. The inhibition of STAT activity by APT and the cross-repression of APT and SLBO conforms to a molecular switch that determines if an epithelial cell becomes motile or remains stationary (Figure 2A). Interestingly, they also observed the resulting two steady states of the model between migratory and stationary cell phenotypes. Although this work was performed on the *Drosophila* model, it illustrates the property of bistability in the JAK/STAT signaling pathway, which is one of the most important pathways in macrophage activation.

Nickaeen et al. developed an ordinary differential equation model to perform bifurcation analysis and revealed mechanisms of macrophage activation and phenotype pattern distribution. They found saddle-node bifurcations in the internal regulators STAT1, STAT6, and NFKB (Figure 2B). LPS was a bifurcation parameter for NF-κB, STAT1, and STAT6 transitions and IFN-γ for STAT1 and STAT6 transitions, while IL-4 levels did not give rise to any bifurcations. These observations confirmed in silico the presence of five switches in macrophage phenotype transitions and enabled different combinations of steady state levels attributable to nine different fates, including M1, M2a, M2b, and other phenotypes, thus describing a multistable system [35]. Furthermore, they developed a model to describe the mutual interactions between a macrophage and its neighboring cells that could affect macrophage fate through cytokine production. Therefore, they implemented an agent-based model, in which each agent represents a cell that is able to interact with other agents. This approach accounts for the stochastic behavior of each single cell, thereby representing the continuous cell population communications. To model these communications, they included IL-4, IL-10, IL-12, and IFN-γ concentration dynamics while introducing a time-scale separation for model simplification to describe cytokine levels in the function of the steady state concentrations of STAT6, STAT1, and NF-κB. In this way, they described a single cell model where the expression levels of STAT1, STAT6, and NF-κB depend on external stimuli and the concentrations of the cytokines can be determined as they diffuse to neighbor cells (Figure 2C). Despite the simplicity of their model, they calibrated it against experimental data and proposed that a dynamic bifurcation is a crucial built-in mechanism of macrophage activation [35]. Since an agent-based model is characterized by stochasticity, the individual agents (macrophages) display different configurations of multistable switches dependent on the corresponding amounts of activating cytokines. Their observation suggests that the extracellular cytokine production influences the control of these multistable switches at the macrophage population level.

While the authors of [35] showed the bistable dynamics of macrophage phenotypes when exposed to external signaling cues, Smith et al. showed that after initial differentiation into M1 and M2, the M2 phenotype was ultimately dominant. They examined macrophage population response to simultaneous or sequential M1 and M2 activation signals to generate a subpopulation dataset based on M1/M2 marker expression, using flow cytometry. They found that M1 treatment potentiates the response to a subsequent M2 treatment, while M2 pretreatment blocks the response to M1 treatment [36]. This is an elegant experimental demonstration of the hysteresis proper to bistable systems, where the state of the system depends on its history [37]. In addition, they observed a heterogeneous distribution of markers, suggesting that the macrophages do not exist in discrete polarized states at the population level.

In addition, their mathematical modeling of candidate regulatory networks indicated that a complex interdependence of M1- and M2-associated pathways underlies macrophage activation. They used six minimal regulatory models of CD86 and CD206 expression in response to the different costimulatory conditions, using ordinary differential equations with different topological motifs. All models were built using generic formulations of self-stimulation and mutual inhibition, which are common building blocks in immune cell differentiation models [38]. Specifically, the authors found that a mutual inhibition motif was by itself not sufficient to reproduce the temporal marker expression data. An incoherent feedforward mechanism of M1 activation as well as both the inhibition and activation of M2 by M1 were required for bistability (Figure 2D). Indeed, they included an additional node called Y to comprise feedback inhibition mechanisms, such as those mediated by SOCS and STAT3 or NF-κB and STAT6 [36]. 

A recent work by Frank et al. also employed bifurcation and sensitivity analysis to reveal the key drivers of multistability in a simple model of macrophage activation, specifically tracking STAT1 and STAT6 activation levels as proxies for M1 and M2 activation, respectively. The authors used ordinary differential equations and included self-stimulation and mutual-inhibition circuits between STAT1 and STAT6 (Figure 2E). They justified this choice because the individual steps are unknown; therefore, they assumed that responses in self-stimulation and inhibition are sigmoidal and can be modeled using a Hill function [39]. Despite its simplicity, the model exhibits complex dynamics. Furthermore, the authors showed that external signaling cues are necessary for macrophage commitment and emergence to a phenotype, but that the intrinsic macrophage pathways are equally important. 

These reports provide experimental and computational evidence of the emergent property of multistability in macrophage phenotypes. They also revealed the underlying principles of those molecular switches that include mutual-activation or mutual-repression circuits (Figure 2). However, realistic biological networks generally encompass more proteins and variables, precluding the use of traditional phase plane analysis to identify bistability in networks with longer mutually inhibitory feedback loops [23]. Thus, current models are very much simplified, precluding the understanding of the exact mechanism of bistability and the identification of targets to interfere with the molecular switch. This explains why larger, more complex models such as that of Zhao et al. were not analyzed for bistability [33].

The reports mentioned above also highlight some of the implications of bistability, particularly hysteresis. For instance, an M2-stimulated macrophage requires much higher concentrations of LPS + IFN-γ to undergo an M1 program compared to a naive, nonstimulated macrophage [36]. They also support the hypothesis that M1/M2 transitions are regulated by multistable pathways and describe interesting properties of this multistability. Although the literature cited in this section tries to extrapolate the multistable behavior of these pathways to macrophage activation, it is obvious that the concept of multistability better applies to molecular pathways and that there is more than one molecular switch at work defining macrophage phenotypes. As was nicely illustrated by Nickaeen et al., five switches led to nine potential different states [35]. Therefore, the higher the number of switches, the higher the number of potential discrete states in macrophage activation, potentially explaining why macrophage activation is considered a continuum at the population level [3], although it is in fact discrete but with many instances defined at the molecular level. 

## 4. miRNA Circuits as Possible Sources of Bistability in Gene Expression

The previous reports indicate that signaling modules involved in M1/M2 programs have the intrinsic potential to give rise to bistability at the signal transduction level. However, many of the molecules reported on above are transcription factors, thus participating in complex gene expression programs interacting with other molecules such as miRNAs, which are small endogenous RNA molecules that bind mRNAs and repress gene expression [40]. As a matter of fact, in order to keep the models simple, no systems biology study of M1/M2 transition bistability has included miRNAs, although they are known to be key regulators in gene expression and gene expression noise [41], especially for low-expressed genes such as transcription factors [42]. A typical miRNA is processed from a long primary RNA sequence into a short mature functional transcript around 22 nucleotides in length. A common characteristic of an miRNA is its ability to pleiotropically target the expression of hundreds or even thousands of genes [43], and their target genes can also be regulated by several miRNAs [44]. Current estimates indicate that the human genome contains 1917 annotated hairpin precursors and 2654 mature sequences of miRNAs [45], estimated to directly regulate >60% of human mRNAs [46]. In consequence, miRNA-transcription networks have a high degree of complexity and there is a high probability that miRNA-transcription factor interactions regulate important targets in M1/M2 transitions. 

An important role of miRNAs in modulating macrophage phenotypic activation is demonstrated by accumulating evidence in which an excessive or impaired inflammatory response of macrophages is found to be tightly linked to the deregulation of miRNAs [38]. For example, some functional miRNAs such as miR-146, miR-125b, miR-155, and miR-9 have been reported to be induced by inflammatory stimuli to attenuate TLR4/IL-1R signaling pathways in monocytes and macrophages [47,48,49]. A wide range of miRNAs regulating the inflammatory profile of macrophages has been identified, including M1-related miRNAs (miR-9, miR-17, miR-20a, miR-98, miR-106a, miR-125b, miR-127, miR-146, miR-147 miR-155, miR-181, miR-451, and miR-720) and M2-related miRNAs (let-7b, let-7c, miR-21, miR-23a, miR-23b, miR-27a, miR-34a, miR-92a, miR-124, miR-125a, miR-132, miR-142, miR-223, and miR-511). We refer the interested readers to the excellent recent reviews [47,50,51]. 

In particular, the modulation of macrophage activation by miRNAs has gained a lot of attention. The overexpression or depletion of miR-155 drove macrophages to the M1 or M2 phenotype, respectively, confirming that miR-155 plays a central role in regulating the serine/threonine kinase (Akt)-dependent M1/M2 activation of macrophages [52], and tumor-associated macrophages were successfully reprogrammed into pro-inflammatory M1 macrophages by miR-155 overexpression [53,54]. The manipulation of miRNAs to regulate macrophage activation has been suggested for gliomas, specifically for miR-142 [55]. Additionally, miRNAs contribute to trained immunity in macrophages due to the long half-life of miRNAs and the limited proliferative ability of macrophages [56]. The upregulation of miR-155 in response to inflammatory signals is associated with macrophage hyperactivation, indicating that cells with sustained miR-155 remain primed in a hyper-sensitive state to increase the response to a secondary stimulus [57]. 

The list of miRNA targets on the pathways involved in M1/M2 transitions is expanding, indicating that they represent strong modulators of M1/M2 transitions, potentially participating in the assembly of a molecular switch circuit. Withstanding with the opposite activity of the M1 and M2 regulators (e.g., STATs) cited above, examples of opposite activities in the miRNAs have also been reported. However, the complexity of this regulatory network is such that the identification of the master key regulators requires the aid of computational approaches. 

Lu et al. correlated miRNA and mRNA expression over time to elucidate the expression profiles of miRNAs and their potential targetomes during mouse macrophage activation. They hypothesized that miRNAs mediate the early events of the M1/M2 phenotype switch through a complex and dynamic miRNA-targeted mRNA interactome network. Their bioinformatic analysis revealed 31 differentially expressed miRNAs, including four top M1 miRNAs (miR-155-3p, miR-155-5p, miR-147-3p, and miR-9-5p) and four top M2 miRNAs (miR-27a-5p, let-7c-1-3p, miR-23a-5p, and miR-23b-5p), which could be divided into an early and a late cluster of miRNA expression. Their integrative analysis of miRNAs and mRNAs demonstrates that the miRNAs regulate nearly 4000 differentially expressed genes and most of the biological pathways that are enriched in macrophage activation [58].

Taken together, the literature includes many potential regulators of M1/M2 transitions, including important transcription factors and many miRNAs, leading to a paramount complexity. In fact, the endogenous transcription networks of miRNA–TF interactions have been reported to assemble complex motifs, including negative feedback loops, positive feedback loops, coherent feedforward loops, incoherent feedforward loops, miRNA clusters, and target hubs leading to nonlinear, systems-level properties such as bistability, ultrasensitivity, and oscillations [59,60]. This means that a differential expression analysis of target genes or miRNAs or the study of the network topology could be insufficient to identify the underlying mechanisms of bistability. Indeed, Cinquin et al. showed ways to derive structural (related to the interaction graph) and numerical (related to the magnitudes of the interactions) constraints required for bistability [22].

Therefore, we suggest that, in addition to bistability arising from the interactions of molecules in the signaling modules of M1/M2 transitions (Figure 3A), these transcription factors also participate in complex circuits with miRNAs, leading to bistability at the gene-expression level. To explore this possibility, we constructed a network of experimentally validated interactions from several databases using the miRNAs and transcription factors cited above and reported to be involved in macrophage activation. We constructed the network using our recently published biocomputational platform BioNetUCR [61]. The resulting network is highly complex, including 148 transcription factors, 24 miRNAs, and 537 genes (not shown). Due to the recent advances in the role of immunometabolism for macrophage activation (see below), we focused on the potential effects of this gene expression network on macrophage metabolism. For this purpose, we filtered the target genes identified in our network through the list of 3696 genes of the human metabolism extracted from Recon3 [62]. The gene expression subnetwork controlling metabolism includes 148 transcription factors, 20 miRNAs, and 105 metabolic genes (Figure 3B). This network shows a central regulatory core of complex interactions and future work is required to assess their potential for the emergence of bistability. In addition, we highlighted the interactions of the mutually-inhibiting regulators of M1/M2 macrophage transitions with the metabolic genes, including both common and exclusive regulatory interactions.

Thus, macrophage metabolism implies an additional layer of complexity and it is very likely that the bistability at the signaling and gene-expression levels directly impacts metabolic genes, explaining the sharp differences in several metabolic pathways for the macrophage phenotypes. Moreover, the differentially activated metabolic pathways could lead to the production of specific metabolites that can be sensed by the same macrophages (or neighboring cells) modulating gene expression (see below), thereby increasing the complexity of the system.

## 5. Metabolic Switches in Macrophage Phenotypes Suggest Bistability in Metabolic Gene Expression

The concept of “immunometabolism” was introduced to describe how metabolic adaptations not only provide the required energy to support immune activity in specific contexts but also directly affect the functions of immune cells via the control of transcriptional and post-transcriptional events [8]. As mentioned above, M1/M2 macrophage activation includes well-coordinated changes in signaling events and post-translational mechanisms with an extensive remodeling of metabolism. The basic phenotypes M1 and M2 describe two possible answers to external stimuli, and their physiological behavior is linked to changes in macrophage metabolic profiles. As we will describe below, this involves the core energy metabolism and the use of nitrogen in particular.

In 1970, Hard demonstrated that M1 macrophages presented an increased glycolytic pathway together with a decrease in oxygen utilization [63], a fact further demonstrated in activated macrophages after an inflammatory stimulus [64]. Subsequent studies have shown the use of glycolysis and the pentose phosphate pathway (PPP) to supply energy in pro-inflammatory phenotypes or M1, as well as an impaired tricarboxylic acid (TCA) cycle, a reduction in fatty acid oxidation, and oxidative phosphorylation (OXPHOS) [65,66]. On the other hand, the metabolic behavior of the M2 phenotype is characterized by a fully functional TCA cycle and energy is obtained from oxidative processes [65,67].

As stated above, the differential gene expression potentially leads to differential metabolic pathway activation and the production of differential metabolites. Bordbar et al. applied flux balance analysis (FBA) for the mathematical modeling of macrophage metabolism using metabolic reconstruction based on gene expression and proteomic data, with the aim of identifying the switches in macrophage metabolism. They compared the rates of biomass, ATP, and NO production with experimental data between M1 and M2 phenotypes and performed deletion analysis to find the essential reactions of M1 and M2 activation on the metabolic functions of ATP, redox (NADH), NO, and extracellular matrix precursors (proline), while maximizing for M1 (NO production) and M2 (proline and putrescine) phenotypes. Out of 76 metabolic subsystems in the network, only 4 had a large difference in response to reaction deletions: alanine and aspartate metabolism; arginine and proline metabolism; OXPHOS; and urea cycle/amino group metabolism. Outside of the reactions that were directly involved in the production of these metabolites, the blockade of OXPHOS reduced proline and putrescine production (M2) but had no effect on NO production (M1). This indicates that OXPHOS is more important for the M2 activation phenotype than for the M1, as expected. Notwithstanding, the blockade of glycolytic reactions had a stronger effect on M1-related NO production than for M2-related proline and putrescine [68]. These results are consistent with a study reporting that M1 macrophages have a higher glycolytic activity whereas M2 macrophages are more dependent on OXPHOS [67].

Hörhold et al. continued the work with the same model as Bordbar et al. to investigate the metabolic switches in gene regulatory and metabolic networks during macrophage activation. Their differential gene expression analysis showed that M1 macrophages activated processes such as inflammation, cell death, proinflammatory cytokine expression, TNF signaling, and specific metabolic processes such as NO synthesis and glycolysis. Notably, the entire biochemical pathway from glucose to lactate production was upregulated, enabling fast energy supply, which is needed for cytokine production and the effective eradication of invading pathogens. On the other hand, M2-like macrophages upregulated anabolic and cellular maintenance processes, including nucleotide and amino acid synthesis. The entire pathway for inosine monophosphate (IMP) biosynthesis, which is the rate-limiting step in purine biosynthesis, is upregulated in the M2 macrophages. Their constraint-based modeling using FBA supported this duality between glycolysis and nucleotide biosynthesis and revealed the first switchlike behavior in the metabolic fluxes: in M1 macrophages they observed higher fluxes of glycolysis and lactate production, whereas M2 macrophages showed a higher flux in the biosynthesis of IMP. A second switch was found for nitrogen, which was either metabolized into nitric oxide as a proinflammatory signaling molecule (in M1 macrophages) or degraded into urea by arginase (in M2 macrophages) [69]. 

Now, we will review how various transcription factors are responsible for the regulation of these metabolic changes, establishing thereby a connection between the potentially multistable signaling and gene-expression circuits and macrophage metabolism. 

Hypoxia-inducible factor 1 α (HIF1α) has been shown to be crucial in the glycolytic process even under normoxic conditions [70]. The non-oxygen-dependent transcription of HIF1α in macrophages is regulated principally by two signaling pathways: TLR/NF-kB, triggered by inflammatory signals [71], and Akt/mammalian target of rapamycin (mTOR), activated by growth factors [72,73,74]. In the latter route, macrophage activation is apparently regulated by Akt kinases in an isoform-specific way, where Akt1 deletion promotes the M1 profile, whilst deletion of Akt2 promotes the M2 phenotype [75]. 

HIF1α also promotes the expression of glycolytic enzymes, glucose transporter GLUT1, lactate dehydrogenase (LDH), pyruvate dehydrogenase kinase 1 (PDK1), and inflammatory mediators [70,71,76,77,78]. LDH and PDK1 prevent the formation of acetyl-CoA from pyruvate and its further oxidation in the TCA cycle, empowering the glycolytic pathway in M1 macrophages. Similarly, M1 macrophages predominantly express a less-active isoform of the glycolysis regulatory enzyme: 6-phosphofructo-2-kinase B, which increases the use of glucose in this pathway [77]. M1 phenotypes also promote the synthesis of isoform 2 of pyruvate kinase, which strengthens its state of activation through the interaction with HIF1α and the regulation of genes under its influence [77,79]. Likewise, some modifications have also been observed in the enzymatic expression of the PPP, for example, a decrease in the expression of the enzyme sedoheptulokinase [65], in such a way that some studies have shown that its overexpression leads to defects in the activation and inflammatory response of M1 macrophages [80,81]. The role of the glycolytic pathway in M2 phenotypes has not yet been clearly understood. Whilst glycolysis has been suggested as an active pathway in M2 macrophages, whose inhibition might interfere with their activation [82,83], recent studies have revealed that these cells can also use glutamine to maintain a functional OXPHOS, as well as their specific physiological functions, even in the absence of glycolysis [84].

One of the essential physiological characteristics of M1 macrophages is their ability to produce ROS, a key element in the physiological activity of these cells, regulating bacteria killing, phagocytosis, and activation as well as modulating the activity of transcription factors such as TRAF6 and mTOR [85,86,87]. Activation directly influences macrophage capacity to generate altered oxygen-based molecules. For example, during the normal process of cellular respiration, the ROS levels produced in the electron transport chain (ETC) are kept low and under strict control by specific enzymatic pathways. In contrast, in cellular states characterized by the reduction of the respiratory chain, ROS production is higher due to an increase in electron leakage and their reaction with molecular oxygen in the mitochondria microenvironment [88,89]. The electron flux through the mitochondrial complexes I and III has traditionally been considered the main process of mitochondrial ROS production, but recent data show that in the presence of an impaired OXPHOS the production of ROS is predominantly associated with a reverse electron transport in complex I [90]. 

Because of their phagocytic activity, macrophages have a high turnover of the plasma membrane; hence, fatty acid synthesis is a key process for these immune cells [91]. Lipids’ metabolism is also fundamental in macrophages’ physiology and their polarization. It has been observed that the transcriptional regulation of lipid metabolism is tightly controlled by sterol receptor element-binding protein (SREBP) and liver X receptor (LXR). In macrophages, it has been shown that SREBP-1a and LXR are overexpressed and that they are involved in regulating cell behavior [92,93]. For example, in LPS-treated macrophages, a NF-κB-mediated increase in the SREBP-1a activity has been observed, whereas inflammasome activation failed in defective SREBP-1a M1 macrophages [94,95]. On the other hand, the activation of LXR leads to a reduction in the proinflammatory response of M1 macrophages mediated by the inactivation of AP-1 and NF-κB [96]. 

The regulation of fatty acid synthesis (FAS) and fatty acid oxidation (FAO) drive macrophage M1/M2 activation, respectively. While FAS is an elemental pathway for energy production and prostaglandin biosynthesis in M1 macrophages [97], the metabolism of M2 macrophages depends on the uptake of fatty acids and their oxidation, a process conducted by STAT6 and the PPARγ [98]. This has been supported by the experimental hindering of fatty-acid transportation to the mitochondria, with the consequent diminution of FAO, which has shown a reduction in M2 polarization [99,100].

In altered inflammatory or immune states, macrophages can adapt to sudden changes in the availability of nutritional sources. Amino acid catabolism has been demonstrated as a central pathway for maintaining the immune activity of these cells. For example, inducible nitric oxide nitric synthase (iNOS) is overexpressed as a response to pro-inflammatory stimuli such as LPS, leading to an increase in the levels of citrulline and NO. The reaction of NO with molecular oxygen and ROS lead to the synthesis of a vast array of reactive nitrogen species and other chemicals with antimicrobial properties [101]. Additionally, it has been shown that NO induction blocks the M1-to-M2 switch, whereas M1 macrophages can be reprogrammed into M2 by the inhibition of iNOS [102]. In contrast, in M2 phenotypes, arginase 1 is overexpressed, and one of its products, ornithine, is transformed into polyamines: putrescine, spermidine, and spermine by the ornithine decarboxylase (ODC). These polyamines are involved in the cell growth and tissue repair characteristic of M2 cell activities. ODC has also been implicated in some chromatin modifications that restrict M1 macrophage antimicrobial activities [103].

To summarize, the current biochemical findings and mathematical modeling of macrophage metabolism confirm a switchlike behavior in macrophage activation controlled by several signaling pathways and transcription factors, including TLR/NF-kB, HIF1α, STAT6, PPARγ, AKT/mTOR, SREBP, and LXR, integrating thereby the multistable pathways observed for the signaling and gene-expression modules mentioned above with macrophage metabolism (Figure 3C). Nevertheless, these differences also lead to the production of differential metabolites with important regulatory functions (see below).

## 6. Metabolite Sensing to Regulate Metabolism or Gene Expression

The metabolites differentially produced by M1- or M2-activated macrophages can theoretically be sensed by the same, secreting macrophages or by neighboring cells at different levels. The question at this point is whether these metabolites provide a positive or negative feedback signal to the corresponding signaling, gene expression, or metabolic modules, either promoting or counteracting the respective macrophage phenotypes. We hypothesize that they are sensed at two different levels. First, the metabolites could modulate metabolic reactions in a paracrine and/or autocrine way by exerting a direct regulatory activity on metabolic enzymes. Second, they could modulate these reactions via specialized sensors for metabolites that alter gene expression programs through signal transduction. We will review the most relevant instances for these metabolites differentially expressed in M1- and M2-activated macrophages with the aim of introducing a novel concept of how the crosstalk between metabolic switches and signaling/gene expression modules confers stability on the corresponding steady states in macrophage phenotypes. 

First, the work by Bordbar et al. offers an interesting example where the metabolites are directly sensed within the metabolic pathways [68]. They determined important metabolic immunomodulators of macrophage activation by sensitivity analysis on the metabolite exchanges of the murine macrophage cell line RAW network for each of the five activation-based objective functions. In general, sensitivity scores for NO, proline, and putrescine did not vary significantly, as their respective productions are coupled with the arginine fate and their synthesis pathways are proximal to one another. They found that oxygen, glucose, and glutamine have a great impact on macrophage metabolism because they are keys to cellular respiration, ATP production, and respiratory burst. Glutamine is required for the biosynthesis of arginine and nitrite/urea, arginine is critical for NO production (M1), and L-cystine was also found to selectively activate NO production. In contrast, valine and isoleucine are also important for ATP, NADH, proline, and putrescine production (M2) but not for NO production. The catabolism of tryptophan and phenylalanine had a suppressive effect on both M1 and M2 metabolic activation phenotypes. For instance, indoleamine 2,3-dioxygenase (IDO) has been reported to inhibit T-cell proliferation in mammals [104] and the tryptophan catabolite 3-hydroxyanthranilic acid inhibits the expression of NO synthase [105]. 

Metabolite production had a suppressive effect on M1/M2 metabolic phenotypes, especially vitamin D3 production, which has also been reported to be a critical negative feedback mechanism to control activated macrophages [106]. Likewise, several nucleotides and deoxynucleotides inhibit macrophage activation, as reported for adenosine [107]. Moreover, hyaluronan production was one of the most suppressive factors of macrophage activation. Other metabolites have a positive effect on activation, including urea, ammonia, and glutamate, although urea was reported to inhibit NO production [108]. Indeed, the inhibition of glutaminase was reported to mediate activated M1-like immune responses in macrophages by increasing glutamate levels [109]. In this study, they only found a few potential differential metabolic regulators directly impacting the metabolic pathways. M2-related proline and putrescine production was more sensitive to the amino acids valine, leucine, and isoleucine compared to NO production. In contrast, L-cysteine, glutamate, and NO itself had stronger effects on M1-related NO production [68]. This indicates that other sensors may be at work to sense the metabolic environment and induce changes in the activation phenotypes of macrophages. 

Second, there are also specialized sensors for metabolites. Currently, the AMP-activated protein kinase (AMPK) and mTOR signaling are among the best-understood metabolite-sensing and signaling pathways. To interact with the environment and coordinate the biological network within, cells need a timely and accurate perception of the dynamic changes in intra- and extracellular metabolites, particularly the concentration of nutrients [110]. We will review the specific sensors for the monitoring of energy and the metabolic microenvironment relevant to macrophage activation. 

As already mentioned, M1 macrophages differ (compared to M2) in the metabolic pathways aimed at obtaining energy from nutrients and the need for specific intermediaries to supply their physiological demands. The mainly glycolytic metabolism of M1 macrophages allows the maintenance of their functionality despite the several interruptions observed in the TCA cycle [111,112,113]. When the TCA cycle’s catabolic function is not required, intermediates such as citrate, itaconate, and succinate are made available to reach the cytoplasm, where they carry out regulatory actions. TCA cycle metabolites can be sensed by their covalent binding to proteins. 

Once in the cytoplasm, citrate takes part in a broad spectrum of regulatory activities, including stimulating lipid synthesis and gluconeogenesis and the production of NADPH and acetyl-CoA, which is used for histone acetylation [64,114,115]. Particularly, in M1 macrophages, the expression of proteins, such as NF-κB, IL-6, and IL-10, is modified by acetylation [116,117]. This cytosolic citrate accumulation is mainly due to both the upregulation of the mitochondrial citrate carrier (CIC) and the downregulation of isocitrate dehydrogenase [65,113], a process responsible for the first interruption of the TCA cycle observed in M1. Experimental studies have shown that the overexpression of CIC is stimulated by LPS, TNF-α, or IFN-γ and that cytosolic citrate is essential in the production of NO, ROS, and prostaglandins E2 [113,118].

Itaconate, a metabolite known for its antibacterial properties [119], is produced at another breaking point of the TCA cycle. It is produced in activated macrophages from cis-aconitate due to a substantial overexpression of ACOD1 (coding cis-aconitate decarboxylase) [120,121]. Itaconate inhibits succinate dehydrogenase, which leads to an accumulation of succinate [66,122]. 

High concentrations of succinate have been associated with a decrease in some cellular processes, such as mitochondrial respiration, ROS production, and proinflammatory cytokine release, as well as the activation of the inflammasome [123]. Succinate signals to its target proteins by the succinylation of lysine residues and coordinates the TCA cycle to enhance cellular antioxidant defense [124] and impair mitochondria respiration [125] and peroxysomal function to increase the production of ROS [126]. Once in the cytosol, succinate enhances HIF1α activity by blocking prolyl hydroxylase. HIF1α stabilization activates the transcription of glycolytic genes [127], thus sustaining the glycolytic metabolism of M1 macrophages. In addition, during inflammation stages, succinate is released by macrophages to the extracellular microenvironment [128], where it can reach succinate receptor 1 [129]. It has been observed in LPS-stimulated macrophages that succinate can amplify the inflammatory response through autocrine signaling [130].

In contrast, M2 macrophages, which are more dependent on OXPHOS [67], show an intact TCA cycle that provides the substrates for the complexes of the ETC. An interesting example of M2 macrophages are tumor-associated macrophages, which are exposed to low oxygen, low glucose, and low ATP concentrations, favoring AMPK activation and promoting an anti-inflammatory phenotype [131]. Indeed, the glucose and oxygen levels can be sensed by the binding of a sugar group to proteins by enzymes such as O-linked GlcNAc transferase. Upon hypoxia, this enzyme promotes the glycosylation of glucose-6-phosphate dehydrogenase, the rate-limiting enzyme for oxidative pentose phosphate pathway [132]. This in turn increases its activity, promoting the anabolism of nucleotides and lipids [133]. 

Moreover, AMPK has the ability to sense the AMP:ATP ratio, as its γ subunit contains binding sites for AMP, ADP, and ATP. When cells have an insufficient energy supply, AMPK is activated to promote catabolism to provide more energy, and anabolism is slowed down to avoid an overdraft of the energy currency ATP [134]. The glucose levels are also sensed by an AMP/ATP-independent mechanism mediated by the glycolytic enzyme aldolase that binds to fructose-1,6-biphosphate (FBP), an intermediate of glycolysis. Upon glucose depletion, FBP-unbound aldolase promotes the association of AMPK with v-ATPase, ragulator, axin, and liver kinase B1, thus triggering AMPK activation [135]. Indeed, it was demonstrated that AMPK is a potent counter-regulator of inflammatory signaling pathways in macrophages, as their stimulation with anti-inflammatory cytokines (i.e., IL-10 and TGFβ) resulted in the rapid phosphorylation/activation of AMPK, whereas the stimulation of macrophages with a proinflammatory stimulus (LPS) resulted in AMPK inactivation. Interestingly, an active AMPK also results in decreased TNFα and IL-6 production upon inflammatory stimuli and increases IL-10 production [136]. This could promote a self-sustaining vicious cycle towards an anti-inflammatory phenotype in tumor-associated macrophages, offering a possible explanation for the stability of the steady state associated with the M2 phenotype.

IDO is the limiting enzyme in the tryptophan catabolism of immune cells. IDO catalyzes kynurenine synthesis and is overexpressed in M2 phenotypes, whereas its silencing induces an M1-like profile [137,138]. Glutamine is also involved in several biosynthetic pathways implicated in the cell proliferation and physiological activities of immune cells [64]. Different routes of glutamine metabolism can reflect M1/M2 activation in macrophages [68]. In M1 macrophages, succinate synthesis is promoted by feeding the TCA cycle with glutamine [127]. On the other hand, glutamine metabolism is able to promote M2 activation in different ways: the stimulation of OXPHOS and FAO, epigenetic reprogramming [139], the downregulation of HIF1α, and the stimulation of UDP-GlcNAc synthesis, which is essential for the abundant glycosylation observed in M2 [65]. Glutamine synthetase is also highly expressed in M2 phenotypes and its inhibition induces an M1-like phenotype after IL-10 stimulation [140].

As reviewed above, M1 and M2 macrophages have important differences in amino acid metabolism. As protein synthesis is a highly energy-demanding process, the energy-sensing AMPK signaling pathway is connected to mTOR, a central cell-growth regulator that integrates growth factor and nutrient signals [141]. Although mTOR itself is not an amino acid sensor, it reads the abundance of amino acids through complexing with specific sensor proteins. With the help of two arginine sensor proteins, SLC38A9 and CASTOR1, the arginine level controls mTOR activity. SLC38A9 is a lysosomal arginine transporter that promotes the formation of an active mTOR complex in the presence of arginine [142]. In contrast, CASTOR1 functions as an arginine sensor inhibiting mTORC1 in a shortage of arginine [143]. The leucine sensing of mTORC1 is performed in a similar fashion, with Sestrin2 as the leucine sensor. Leucine-bound Sestrin2 is released from GATOR2, leading to mTOR Complex 1 (mTORC1) activation in leucine sufficiency [144]. Leucin binds to Ras-related GTP-binding protein A/B (RagA/B), another key component in the mTORC1 complex, also in the abundance of leucine. This differential binding of amino acids to proteins is called aminoacylation, which is tRNA-dependent and is specifically restricted to arginine, leucine, and phenylalanine [145]. A mass spectrometry study revealed 43 argininylated proteins that were involved in actin cytoskeleton, actin binding, calcium binding, microtubules, signal transduction, acetyl-CoA metabolism, protein biosynthesis, protein folding, protein transport, glycolysis, the regulation of development and transcription, protein degradation, ubiquitin pathway, nucleotide biosynthesis, DNA binding, chromatin remodeling, oxygen transport, and immune response [146]. This suggests that the amino acid modification of proteins serves as an important mechanism of transmitting amino acid signals into biological networks that include mTOR. 

Furthermore, mTOR is also a very critical factor in macrophage activation. A study reported that M1 activation in the livers of mice infected with the highly virulent *Ixodes ovatus ehrlichia* is dependent on mTORC1 activation. In this model, the blockade of mTORC1 activation with rapamycin decreased the frequency of Th17 and enhanced autophagy, which in turn decreased inflammation as well as the pathogenic immune response. In addition, in a mouse model of constitutive mTORC1 activation, the authors observed that the macrophages are refractory to IL-4-induced M2 activation but produce increased inflammatory responses to proinflammatory stimuli [147]. Indeed, mTORC2 signaling is required for the generation of M2 macrophages, while the deletion of mTORC1 signaling in C57BL/6 mouse macrophages led to enhanced M1 macrophage function in vitro and in vivo, despite a significant defect in M1 macrophage glycolytic metabolism [148]. These findings highlight the key role of mTOR in macrophage activation as sensor of the metabolic microenvironment able to control the same metabolic status and provide stability to macrophage phenotypes. This suggests that mTOR represents an interesting target to control macrophage activation [149].

These observations suggest that M1 macrophages contribute to their metabolic microenvironment with metabolites that return a positive feedback signal to the signaling, gene-expression, and metabolic modules. This includes glutamine, glutamate, NO, citrate, itaconate, and succinate promoting the maintenance of the M1 phenotype, with only vitamin D3 reported to exert a negative feedback signal. In contrast, M2 macrophages produce other metabolites such as urea, valine, leucine, isoleucine, and arginine that self-perpetuate an M2 phenotype. These signals are sensed directly within the metabolic pathways or by specialized sensors coupled to the two master regulators of cell metabolism, AMPK and mTOR, to maintain the respective phenotypes. This suggests that macrophage metabolism contributes to the stability of the associated steady states and therefore could be a target to trigger the bistable switch in one or the other direction of macrophage activation. Acetyl-CoA, NAD+, alpha-ketoglutarate, and succinate alter histone acetylation, thereby changing gene expression to alter trained immunity programs (for a review, see [2]). In addition, the shift from oxidative phosphorylation towards glycolysis was found to be dependent on the Akt/mTOR/HIF-1α pathway and has been reported to be essential for trained immunity induced by β–glucan [72].

The identification of the most robust sources of stability could lead to the discovery of possible ultrasensitive targets to perturb the hysteretic behavior of these molecular switches. For instance, the sensing of differentially produced metabolites in M1 and M2 macrophages seem to converge in AMPK and mTOR, suggesting that these two molecules represent potential hubs controlling bistability by establishing a complex, large-scale positive feedback circuit involving signaling, gene expression, and metabolism.

## 7. Modulating Metabolism to Reprogram Macrophages

Two types of intervention are considered to trigger the bistable switch in macrophage activation by modulating metabolism: a direct way, by targeting macrophage metabolism, and an indirect way, by targeting key transcription factors. The latter approach necessitates identifying these key factors. 

Hörhold et al. employed linear regression models to predict potential transcription factor-regulating genes with differential expression among the selected biochemical pathways in which they observed metabolic switches, including glycolysis, the citrate cycle, the pentose phosphate pathway, fatty-acid metabolism, IMP synthesis, and arginine biosynthesis. For this purpose, they used binding information from ChIP databases to optimize a model able to estimate the gene expression profiles and thereby identify the transcription factors that were most often selected by the models across all target genes and construct a fitted parsimony model. Using this model, they performed the in silico reprogramming of M2-like macrophages into M1-like macrophages by replacing the activities of the predicted regulators in M2-like macrophages with their activities in M1-like macrophages. Thereby, they identified five transcription factors enabling the reprogramming of M2-like macrophages into M1-like macrophages by silencing. The majority of the relevant genes were regulated by MYC, followed by E2F1, CTCF, PPARγ, and STAT6. They validated their model predictions using the transfection of siRNA pools targeting the four transcription factors E2f1, Myc, Pparγ, and Stat6 and observed that 66% of the genes were correctly reprogrammed towards an M1-like phenotype. This observation was supported by a partial shift in cytokine secretion [69]. Their elegant approach confirms that the metabolic switch observed in macrophages is controlled by a bistable set of transcription factors undergoing a switchlike behavior.

Another approach to alter macrophage phenotype is the direct targeting of their metabolism. It has been shown that metabolism-related proteins (e.g., sedoheptulokinase) or the concentration of metabolites (e.g., α-ketoglutarate) directly control the inflammatory phenotype of macrophages [81,139]. Moreover, as explained above, the mitochondrial activity is an important factor in macrophage activity, which we suggest could act as a bistable switch. Suggesting that the maintenance of energy production and metabolic homeostasis is a fundamental (if not the most important) basis for cell survival, the important role of mitochondria for the cell fate is easy to imagine. Importantly, Van den Bossche et al. identified that the inhibition of mitochondrial OXPHOS prevents a switch from M1 macrophages to M2 macrophages [102], demonstrating its high relevance for macrophage immune functions. Although it is not yet clear how exactly mitochondria and macrophage activation are regulated by each other, their tight connection highlights mitochondria as a therapeutic target to alter the phenotype of macrophages. The indirect targeting of mitochondrial activity could be achieved by targeting the concentrations of metabolites used in the TCA cycle (e.g., glutamate). Recent data, including our own, suggest a connection between the glutamate metabolism and the inflammatory status of macrophages, highlighting its possible use as therapeutic target [150,151]. Another approach would be the direct targeting of mitochondrial function, as shown by Geiß at al. [152]. Using primary human macrophages, we could demonstrate that the mitochondrial integrity is a major regulator of macrophage phenotype. Probably due to its high relevance for cell survival, these signals can even outperform extracellular stimuli. This strong influence of cell metabolism on macrophage immune function also demonstrates its value for therapeutic uses. This aspect is not part of this review, but has been recently summarized in [153,154].

Furthermore, there is evidence that the generation of ROS in particular, which is mainly produced by the mitochondrial ETC, is a leading force of macrophage activation. Studies report that, e.g., the natural compound curcumin or the diabetes drug metformin can alter macrophage phenotypes, probably by stabilizing the functionality of mitochondria (for a detailed review, please see [155]). 

## 8. Conclusions

In this review, we presented compelling elements from in vitro and in silico studies that support multistability (more than one steady state) or bistability (a specific case of multistability for a system consisting of two steady states) in the pathways regulating M1/M2 transitions in macrophages. We observed and described the switchlike behavior of these transitions at four different levels of integrated cellular programs. We identified possible hubs of bistability as potential targets to control macrophage phenotype transitions through the modulation of the metabolic environment. The regulation of this metabolic microenvironment indeed appears to be a promising strategy to revert or block macrophage transitions.

The first level is that of the signal-transduction pathways related to M1/M2 response to cytokines (Th1- or Th2-derived) or other pro-inflammatory stimuli (e.g., LPS) which represent a first decision-making module (Figure 3A). The circuits established by the STAT family of proteins represent interesting candidates for the robust manipulation of macrophage phenotypes, although the precise mechanisms governing this bistability are incompletely elucidated due to the limitations to the study of this property in larger systems, including the regulation of gene expression.

Second, a switchlike behavior is observed at the gene-expression level. In response to the signaling modules, many of the activated/inhibited molecules are transcription factors that regulate gene expression and form complex networks with other regulators, including miRNAs (Figure 3B). Interestingly, miRNAs are regulators of noise in gene expression [41], especially for low-expressed genes such as transcription factors [42]. An increase in the noise intensity distorts a bistable system into a monostable system [18]. When the noise intensity crosses a critical value, the system loses its bistability and the system becomes monostable. In such conditions, the system cannot be reversed back to the previous steady state, making it irreversible, though the corresponding deterministic system shows reversible hysteresis. The influence of this stochasticity on the reversibility depends on the strength of the noise intensity, which again depends on the feedback parameter value [18]. Therefore, the manipulation of the noise in gene expression by the perturbation of specific miRNA levels such as miR-155 and miR-146 holds great promise to modify the reversibility of steady-state transitions. However, the precise identification of such critical nodes involved in M1/M2 transitions will require the quantitative analysis of mechanistic dynamical models of these interactions by means of sensitivity/stability analysis combined with predictive simulations, including stochasticity. This approach could lead to the identification of processes that can become potential therapeutic targets robustly controlling these molecular switches at the gene-expression level [59,60,156].

Third, these switchlike changes at the gene-expression level impact the expression of genes coding for enzymes or other proteins related to metabolic pathways. This leads to several important switches between M1 and M2 phenotypes in energy production (glycolysis versus OXPHOS), the activation of the PPP to produce NADPH and ROS in M1, a broken TCA cycle to provide succinate in M1 versus a complete TCA cycle to feed OXPHOS and the biosynthesis of nucleotides and amino acids in M2, and the metabolism of nitrogen to NO (M1) or ornithine and urea (M2) (Figure 3C). There is evidence showing how these metabolic pathways can be sensitive to direct manipulation by the same levels of metabolites or substrates, suggesting that the regulation of the metabolic microenvironment could be a promising strategy to revert or block macrophage transitions. As mentioned earlier, potential strategies to regulate macrophage metabolism include the targeting of key transcription factors (such as MYC) or the direct targeting of mitochondria. 

Fourth, these metabolites could also be sensed by specialized metabolic sensors, including transporters and reactions such as succinylation, glycosylation, and aminoacylation, controlling the activity of the two master regulators of cell metabolism: AMPK and mTOR. The current evidence suggests that the M1 metabolites are sensed by the corresponding sensors to maintain an AMPK-off/mTOR-off phenotype that promotes M1 signaling, gene expression, and metabolism. Likewise, M2 metabolites are sensed, leading to an AMPK-on/mTOR-on phenotype that promotes M2 programs and metabolism. These positive feedback loops are self-maintaining activities that could represent a strategy to provide stability to both phenotypes but also suggest that AMPK/mTOR circuits are robust controllers of the bistable switches in macrophage M1/M2 transitions and potential therapeutic targets.

The manipulation of macrophages in order to shift these cells from an immunosuppressive to an immunostimulatory state, or vice versa, shows tremendous potential for treating a broad range of widespread diseases such as sepsis, cancer, or chronic inflammatory diseases. An example of bistable switching in disease proving to be beneficial by altering macrophage phenotypes is AMPK targeting using 5-aminoimidazole-4-carboxamide ribonucleoside or metformin [157,158]. Another example of the successful regulation of the hysteretic behavior of trained immunity is the short-term inhibition of mTOR in organ transplantation, abolishing the need for long-term immunosuppression [159].

Nevertheless, we need to implement integrative systems biology approaches to quantitatively identify the most robust therapeutic targets to turn off/on these switches and direct the cells towards a self-maintaining phenotype of interest. We are convinced that the concept of multistability and its implementation in further studies will tremendously help the identification of these novel targets. The manipulation of multistability represents the most robust type of intervention, since a pulse signal able to switch the steady state will be self-perpetuated over time by the stability of the system at the new steady state.

## Figures and Tables

**Figure 1 cells-11-00404-f001:**
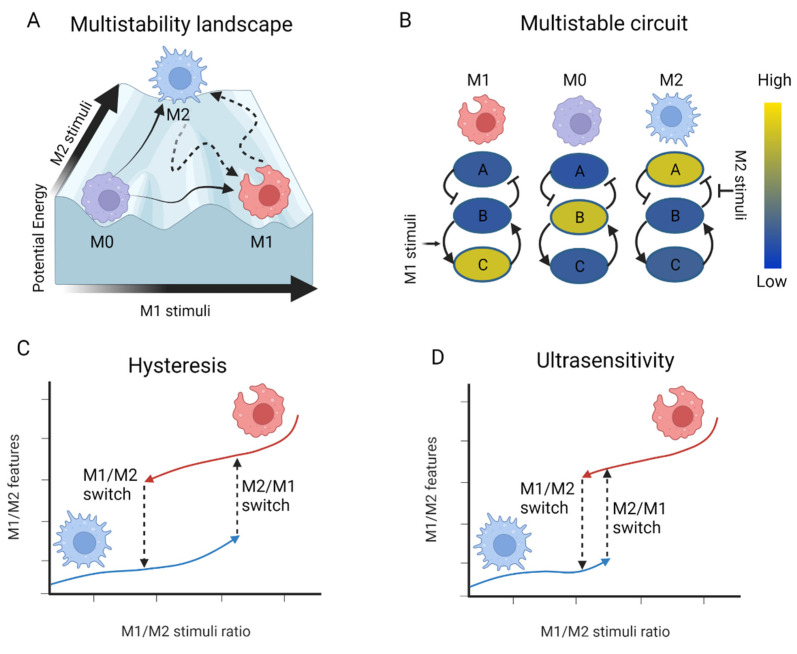
Concept of multistability and hysteresis in macrophage activation: (**A**) Concept of multistability in macrophage programs. In a simplified model the activation landscape of macrophages in vitro leads to two extremes of the full dynamic range of activation. This landscape includes at least three different steady states or attractors with certain levels of stability, the phenotype of the resting (M0), the classical pro-inflammatory (M1), and the alternative anti-inflammatory (M2) macrophages. Different types of signals can direct a resting macrophage out of its attractor zone into a different self-perpetuating steady state (valleys of potential energy), unless the opposing signals reach enough intensity to overcome the stability of the corresponding steady state (slope), leading to a macrophage phenotype transition (switch in steady states). M0/M1 and M0/M2 switches follow specific trajectories (solid lines). Hypothetical M1/M2 transitions are shown with dashed lines. (**B**) The property of multistability typically emerges in biological systems that contain a positive feedback loop (**B**,**C**) or a mutually inhibitory, double-negative feedback loop (**A**,**B**). Here, the 3 potential configurations of this hypothetical circuit are shown for the corresponding attractors, although many other multistable switches could be at work to precisely define a macrophage phenotype (color scale indicates expression levels). (**C**) Concept of hysteresis. The steady state of the systems does not depend only on the current parameters but also on its history. Due to the multistability in macrophage activation pathways, the putative trajectory from an M1 to an M2 phenotype is not the same as for the trajectory from M2 to M1. An M2 macrophage stores an M2 memory or attraction (due to the intrinsic stability of the M2 steady state) and requires a higher amount of M1 signals to undergo a switch to M2 compared to a resting macrophage. Likewise, an M1 macrophage stores an M1 memory or attraction (due to the stability of the M1 steady state), requiring a higher level of M2 signals to undergo a switch to M2 compared to a resting macrophage. The equilibrium points where these switches take place are called bifurcation points. (**D**) The identification of ultrasensitive targets modifies the behavior of these bifurcations, reducing the amount of M1 or M2 signals required to undergo a switch. Figure created with BioRender.com (accessed on 22 December 2021).

**Figure 2 cells-11-00404-f002:**
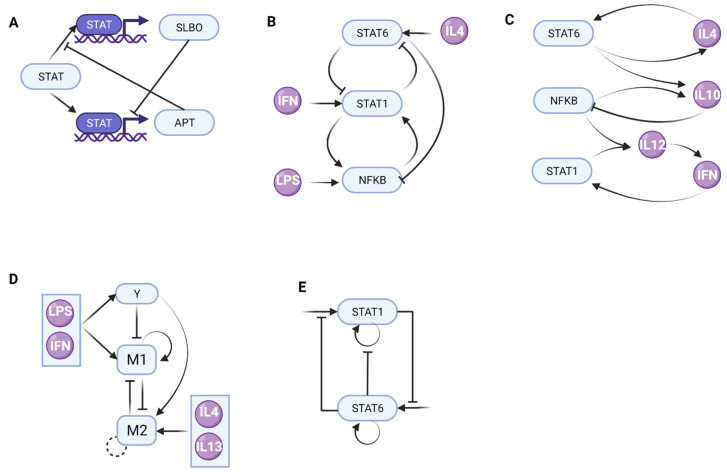
Models showing multistability of signaling pathways involved in macrophage transitions: (**A**) The inhibition of STAT activity by APT and the cross-repression of APT and SLBO conforms to a molecular switch that determines if an epithelial cell becomes motile or remains stationary. (**B**) The mutual inhibition and mutual activation of STAT6, STAT1, and NFKB creates saddle-node bifurcations in macrophage polarization. (**C**) An agent-based model describes macrophage population behavior in the context of cytokine production to influence bistability switches and demonstrates that a dynamic bifurcation is a crucial built-in mechanism of macrophage activation. (**D**) A minimal regulatory model shows that a mutual inhibition motif is not by itself sufficient; an incoherent feedforward mechanism of M1 activation as well as both inhibition and activation of M2 by M1 are required for bistability. Y represents feedback inhibition mechanisms. (**E**) A simple model of macrophage polarization tracking STAT1 and STAT6 activation levels as proxies for M1 and M2 transitions, including self-stimulation and a mutual-inhibition to demonstrate that both external cues and intrinsic pathways are equally important for the emergence of bistability. Figure created with BioRender.com (accessed on 22 December 2021).

**Figure 3 cells-11-00404-f003:**
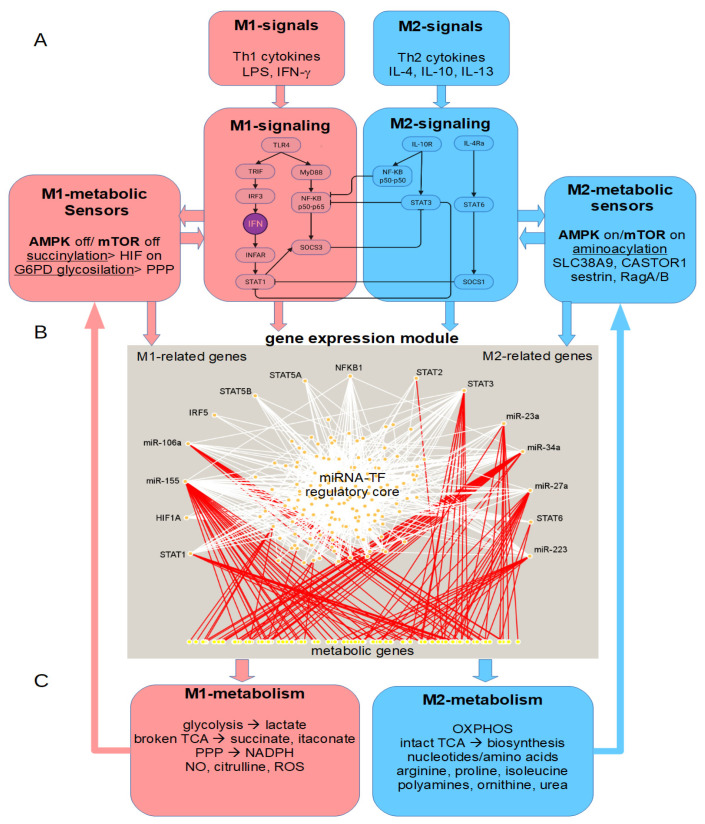
Integrated perspective of the bistability and the switchlike circuits in macrophage M1 and M2 activation: (**A**) M1 and M2 signals activate their respective signaling modules, including mutually inhibitory interactions with potential bistability and the influence of the metabolic master regulators mTOR and AMPK. (**B**) The signaling modules modulate the gene expression of several miRNAs and transcription factors that assemble a complex regulatory core network with the potential emerging property of bistability, including many counter regulators in M1/M2 gene expression. These gene expression modules in turn alter the expression of metabolic genes in a switchlike fashion. (**C**) The differences in M1/M2 macrophage metabolism due to gene expression include contrasting pathways regarding energy production, nitrogen metabolism, TCA cycle, NO, ROS and biosynthesis. The products of this metabolism shape the metabolic microenvironment, which is in turn sensed by the same metabolic pathways or by specialized sensors that report to the master regulators of cell metabolism: mTOR and AMPK (see **A**). These master regulators influence signaling and gene expression to maintain the corresponding phenotypes, conferring stability to the alternative steady states of macrophage activation.

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
