# Peer review of "Multistability in Macrophage Activation Pathways and Metabolic Implications"

_cells, 2022, doi:10.3390/cells11030404_

Round 1

Reviewer 1 Report

In this review the authors appear to be illustrating two main points: that macrophage polarization is controlled by a series of switches that may be manipulated to alter phenotypes resulting from previous stimulation, and that systems level approaches are needed to fully understand the full complement of these switches. The classical M1/M2 paradigm is used to discuss these concepts and 4 types of “switches” are identified including signal transduction, regulators of gene expression, metabolic activity and metabolic sensors. Ultimately, the authors propose that these switches (or “bistable” targets) are key to understanding/treating disease. The general idea is an interesting one, particularly as it may pertain to therapeutic targeting, however the main arguments and concepts are not portrayed clearly which made it difficult to understand the key take-away messages of the review. The below comments are suggestions for the authors to take into consideration to improve the overall clarity

Major comments

  1. My main critique of the review is that concepts are not well defined early in the paper, and existing definitions are bogged down by jargon. It took several reads just to grasp some of the main points that are the foundation of the review. For example, “bi-stability” is only sufficiently defined in the legend of figure 1 but the use of terms such as “self-perpetuating steady state” prevents a clear understanding of what this means in the context of macrophage biology. The concept of bi-stability (and other key terminologies) as it applies to both macrophage phenotypes and/or molecular switches should be explained clearly in the introduction. My understanding is that it is simply the possibility of existing in 1 of 2 mutually exclusive states (i.e. on or off), but it is still not clear from the text. Additionally, it seems that term “bi-stability” is alternately applied to the overall macrophage physiology (M1 vs M2) or specific signalling events in response to stimulation (such as STAT signalling), and which of these the authors are referring to became less clear throughout, which added to the confusion.

  1. Several examples of bistable switches or systems are discussed which possess drawn out details that dilute the key ideas – for instance the complete description of complex signalling cascades within text is not necessary. A specific example is the first paragraph of section 3 in which a large number of signaling proteins are listed, but ultimately the paragraph is very hard to digest and is also made redundant by the summary in the following paragraph which gets the message across clearly without the first paragraph – if the authors would like to highlight all the interactions within the cascade, and their bistable nature, a figure would serve better than a descriptive paragraph. In several places throughout, eliminating many of the specific details, or sticking to one specific clear example within a signaling cascade/transcriptional system/metabolic network, may help to strengthen the main messages.

  1. The authors provide a nicely written, detailed overview of macrophage metabolism, however in these sections the discussion or relevance of bi-stability is somewhat lost. Given the title of the review, identifying and discussing specific targets as bistable switches in the context of their M1/M2 function is needed for illustrating the authors’ notions of bistable regulation and it’s connection to metabolism

  1. The authors illustrate the complexity of molecular decision making required to drive macrophage phenotypes, highlighting 4 major hubs, and suggest that systems level analysis is required to fully unravel the key nodes that can control macrophages stability. Considering all 4 hubs and the authors’ assertions that system analysis will unveil key regulatory points, it would be pertinent to have some speculation/discussion about which specific targets/hubs may be most important for further investigating new macrophage-based therapies as revealed by systems-based studies.

Minor comments

  1. “M2-type responses represent the “resting” phenotype and are observed in healing-type circumstances without infections” – this should be reworded. Macrophages sharing M2-like properties are seen strongly in many types of infection, as well as tumors, damaged tissue and atopic disease. While more similar to “resting” macrophages, many aspects of alternative activation are not seen without stimulus and this varies from tissue to tissue. However in general M1/M2 terminology should be restricted to in vitro models.

  1. The use of the term “steady-state” led to some confusion. Macrophages treated in vitro continue to differentiate and alter their physiology over time from one initial stimulus, which in many cases is also biphasic (i.e. differential timing in cytokine expression, accumulation of suppressive metabolites, etc.), so it’s not entirely clear what the authors mean when they say macrophages exist in different “steady-states” after stimulation

  1. The authors provide a clear in-depth description of hysteresis, but provide no examples of its relevance to the discussed molecular switching in any of the 4 hubs for macrophage activation. Are there examples of hysteresis that can be discussed specifically? This seems like a very important concept, particularly for therapy – i.e. how does infection/disease/treatment history alter macrophage sensitivity to new treatment

  1. In figure 1, it’s unclear what the axes represents (i.e. what is the relevance of “energy”, how are “features” quantitatively defined?). Would be helpful to add labels into the figure to identify the terms represented in A, B and C.

  1. One of the ideas the authors seem to put forth is the notion that these bistable switches can be therapeutically used to convert phenotypically activated/differentiated macrophages. While the main discussion focusses on understanding these switches in vitro, a small paragraph demonstrating specific examples where bistable switching in disease has proven to be, or could be beneficial by altering macrophages phenotypes, would be welcome.

Reviewer 2 Report

This article is about systems biology and specifically discusses the principle of bistability using macrophage polarization as an example with an additional focus on immune metabolism.

The work is comprehensive, very well written and of high quality.

In a first chapter the terms bistability, hysteresis and ultrasensitivity are defined including explanations, how positive feedback loops or mutually inhibitory, double-negative feedback loops regulate the transition from one macrophage activation state to another.

First, current evidence is summarized. In a special chapter, a rather recent aspect is addressed: the role of miRNA in the regulation of bistability in macrophage polarization.

The section “metabolic switches” describes how important metabolic pathways such as glycolysis, pentose phosphate pathway, OXPHOS contribute to bistability. Signaling pathways and transcription factors are also discussed (TLR/NFkB, HIF, STAT6, PPAR-gamma).

In the chapter “metabolite sensing” the authors elaborate on AMPK and mTOR as well as on specific metabolites including citrate, itaconate and succinate.

Finally, the authors discuss the possibility to reprogram macrophages by influencing their metabolism.

Overall, the sections on systems biology on the one hand and those on concrete experimental investigation of macrophage biology on the other hand are well balanced.

The graphs are nice and clear

A detailed summary concludes the article.

I wonder whether the authors may briefly address the concept of trained immunity (Netea MG Science 2016).

Very minor point: line 758: These observations suggests…

Reviewer 3 Report

In this review, the authors present concepts coming from systems biology to analyse macrophage polarization. The authors advocate that the use of a description using the concept of switch-like behavior coule help to better understand the transition that could appear between M1 and M2 macrophages which are identified as the two "stable" states of a bistability pattern in macrophage polarization.

My first remark will be that more than 100 reviews have been publish during the last 5 years with "macrophage polarization" in their title. IT implies that to publish a new review on this theme needs to be associated with either new concepts or the description of original results that have shifted our view on this subject. To this respect, the idea to use systems biology concepts to clarify this field is a good idea and justify to propose a new review.

Despite this promise, the manuscript seems to be finally classical in its presentation. Except sections 2 and 3, this review present a long list of results whitout proposing a new way to interpret them or to use them to cast a new understanding in this field.

The authors do not discuss the fact that the notion of polarization or activation state of macrophages is not a simple concept. What is exaclty a M1 macrophage? Depending on the stimulus, the species, the level of analysis (transcriptomic, proteomic, functionnal (phagocytosis...)) etc... two researchers i nthe field could disagree on the fact ahtat they both work on M1 macrophages. This problem is not only a question of definition, it is also deeply invovled in the misunderstanding  of the functions of particular macrophages like tumour associated macrophages according to a M1/M2 classification.

The authors should discuss this point and use the "consensus" on how we should design the polarization of macrophages (Murray et al, Immunity 2014). The fact that the polarization spectrum of macrophages is a continuum should be discuss especially as it could dampen the enthousiasm of the authors for the bistability concept.

To illustrate this point, I found in this manscript notably in the introduction ambiguous statements about M2 macrophages according to their STAT pathways. Indeed, IL-4 is related to the phosphorylation of STAT6 and IL-10 to STAT3. But these two M2 macrophages are quite different from a fonctionnal perspective for example according to their respective ability to phagocyte apoptotic cells.

Concerning section 2, I have the following remarks:

The concept of emergence is a quite complex story in the field of philosophy of science. Stating that bistability is an emergent concept is not an obvious statement. If you go on this line and pursue the analogy of Figure 1A toward an equivalent mechanical system, having to stable equilibirum position for a a solid moving on this energetic landscape leads to the concept of bistability but nobody will claim that this bistability is emerging from this mechanical system. THe concept of emergence is more appropriate in contexts when the laws of one level of the hierachical organisation of a system are unable to predict new laws at a superior level. Is it the case here, I am not convinced. The bistability could be explained by the dynamical differential equations of feedback loops at the molecular level (like those of Figure 2). Where is the emergence of new laws here? 

The Figure 1 needs some modification.

Strickly speaking as it is described, the energetic landscape of figure 1A could not lead to hysteresis behavior of figure 1B. It is needed to introduce another parameter (so a new dimension) that would modify the potential leading to a catastrophe (in its mathematical meaning) to obtain hysteresis.

The choice of units is completely arbitrary so the values should be removed. It is absolutely not obvious that experimentally the M1/M2 stimuli ratio is the only relevant variable, the absolute values could play a role in such a non linear system.

Putting Energy in the Figuer 1A is ambiguous. Why thermodynamic variable are you speaking about? is this just the enegetic barrier of one molecular reaction whihc one? why introducing real equilibrium states? If we follow this picture it is more energetic to stay M1 if we increase M1 stimulus after the stability value, what is the experimental basis of this statement?

In section 3 the sentence line 267-268 is not clear what are bistability switches at the population level?

My feeling is that the authors wanted to grasp too many results. It would be more profitable to better explain what the bistability concept could bring to the macrophage's biology field even if it implies to enter into more mathematical details about he meaning of states, stimulus than to present a list of miRNA or metabolic pathways if they do not enter into a clear model that could bring new predictions or insights.   

Round 2

Reviewer 3 Report

The authors have provided modifications that take into account my previous remarks. I think that this review could be accepted for publication in Cells.  

Minor point: The reference 21 should be edited as the journal does not appear.